

# Invariants of the Spatial-Energy Structure and Modeling of the Earth's Ion Radiation Belts

## Alexander S. Kovtyukh

Skobeltsyn Institute of Nuclear Physics, Moscow State University, Moscow, 119234, Russia;
kovtyukhas@mail.ru

**Abstract** The spatial-energy distributions of proton fluxes in the Earth's radiation belts (ERB) are well studied and the NASA averaged empirical models constructed for them (the latest versions are AP8 and AP9). These models are widely used in space research. However, for heavier ERB ions (helium, oxygen, etc.), much less measurements were made on satellites, especially in the energy range from tens to hundreds of MeV, and there are no sufficiently complete and reliable models for them. Meanwhile, such ions, although there are much smaller than protons, play a very important role in the physics of ERB, especially in their dynamics, as well as in solving problems of ensuring the safety of space flights. The data on such ions represent a rather fragmentary picture, in which there are significant "white spots". Using the methods considered in this paper, these fragmentary data can be streamlined, linked to each other and get a regular picture that has a simple physical meaning. Spatial-energy distributions of the stationary fluxes of protons, helium ions and ions of the CNO group with energy from 100 keV to 200 MeV at $L \sim 1$–8 considered here on the data of the satellites for 1961–2017. It is found, that results of the measurements of the ion fluxes are arrange in certain regular patterns in the spaces $\{E, L\}$ and $\{L, B/B_0\}$. This effect connected with the existence of invariant parameters of these distributions of ion fluxes. These invariant parameters are very useful and necessary for constructing the ion models of the ERB. The physical mechanisms leading to formation spatial-energy structure of the ERB ion fluxes and the values of its invariant parameters discussed here. In the course of this work, solar-cyclic (11-year) variations in the distributions of helium and carbon-nitrogen-oxygen ions fluxes in the ERB studied for the first time. It shown that, as compared with such variations in the proton fluxes studied earlier, the amplitude of the variations of heavier ions is much larger and increases with increasing their mass.

**Keywords**. Magnetospheric physics (energetic particles trapped)



## 1 Introduction

The Earth's radiation belts (ERB) consist of charged particles with energy from $E \sim 100$ keV to several hundreds of MeV. They trapped by the geomagnetic field at altitudes from $\sim 200$ km to $\sim 50$–70 thousands kilometers. The ERB consist mainly of electrons and protons. In the ERB there are also ions of helium, oxygen, and other elements with the atomic number $Z \geq 2$ ($Z$ is the charge of the atomic nucleus with respect to the charge of the proton). Fluxes of ions and its distributions vary during geomagnetic disturbances. These fluxes depend also on the phase of the solar cycle, conditions in the interplanetary space, and other factors.

Particles with different energy and with different pitch-angles $\alpha$ ($\alpha$ is the angle between the local vector of the magnetic field and vector of the particle velocity) which injected into some point of the geomagnetic trap, drifting with conservation of the invariants $\mu$ and $K$ gradually populate a narrow layer around the Earth, so-called drift shell (Alfvén and Fälthammar, 1963; Northrop, 1963). Therefore, experimental data on the ERB most simply represented in coordinates $\{L, B\}$, where $L$ is a parameter of a drift shell and $B$ is the local induction of a magnetic field (McIlwain, 1961). For a dipole magnetic field parameter $L$ is a distance, in the equatorial plane, from the given magnetic field line to the center of the dipole (in the Earth's radii $R_E$).

Outer and inner regions of the ionic ERBs formed and maintained in dynamic equilibrium with the environment by the various mechanisms (see review Kovtyukh, 2018).

The outer belt ($L > 3$) of ions is formed mainly by the mechanisms of radial diffusion of the particles of the hot plasma from the periphery regions of the magnetosphere into the geomagnetic trap under the action of low-frequency fluctuations of electric and magnetic fields which resonate with drift periods of the trapped particles. This transport accompanied by betatron acceleration of ions, and by ionization losses and charge exchange of ions with atoms of the residual atmosphere.

The inner belt ($L < 2.5$) of protons with $E > 10$ MeV is forming mainly as result of decay of neutrons knocked from the atomic nuclei of the atmosphere by the Galactic Cosmic Rays. For protons with $E < 10$ MeV, this mechanism (CRAND) supplemented by the radial diffusion of particles from the outer to the inner belt. The inner belt of ions with $Z > 4$ formed mainly from ions of the Anomalous component of Cosmic Rays (ACR).

In the intermediate region ($2.5 < L < 3.5$) during strong magnetic storms is operated also the mechanism of capture ions from the Solar Cosmic Rays (see, e.g., Selesnick et al., 2014).

Thus, the main mechanisms of formation of the ERB, sources and losses of these ions we know. However, for the comprehensive verification of the physical models and to identification of the mathematical models parameters it is necessary primarily to create sufficiently complete and reliable empirical models of the ERB for each ion component.

These models one can create only on the base of experimental data obtained over many years and decades. Such models (see, e.g., Ginet et al., 2013) created for protons (AP8/AP9). However, measurements of fluxes of ions with $Z \geq 2$ represent a difficult technical problem, due to the small fluxes of these ions and high background fluxes of protons and electrons. We facing here with the problems of limited and incomplete information. Therefore, sufficiently complete empirical models of the ERB for ions with $Z \geq 2$, similar to models for protons, we do not have, although there are separate fragments of empirical and semi-empirical models for these ions. For example, such models presented for the oxygen ions trapped in the ERB from the ACR (Selesnick et al., 2000; Selesnick, 2001).

When constructing such models, it is necessary to coordinate the data of various experiments. More or less significant discrepancies in the results of these experiments connected with differences in the construction of instruments and in the trajectories of the satellites, differences in the energy ranges and in the angular characteristics of the instruments, differences in the phase of the solar cycle, and with many other physical factors. These factors influence on the fluxes of ions with $Z \geq 2$ in the ERB more significantly than on the proton fluxes (see, e.g., Kovtyukh, 2018).





The creation of the ERB models for ion with $Z \geq 2$ is also necessary in connection with
estimates of the radiation hazard of space flights for humans and with the widespread introduction
of new radiation-sensitive technologies into the space equipment.
This paper consider some approaches to solving the problem of creation of the empirical models
of the ERB for ions with $Z \geq 2$ and presents the first model versions of the stationary ERB for
helium ions and for ions of the CNO group in the range $E \sim 0.1$–100 MeV/nucleon at $L \sim 2$–6.
In the following sections, we consider the experimental data on the spatial-energy structure of
the ERB for protons, helium ions and ions of the CNO group. We consider invariants of the
structure of the ERB (Sect. 2), distributions of ion fluxes in the spaces $\{E, L\}$ and $\{L, B/B_0\}$ and
solar-cyclic variations of these distributions (Sect. 3), physical mechanisms of these structure
formation and the values of its invariant parameters (Sect. 4). Section 5 concludes the paper.

## 2 Invariants of the ion Earth's radiation belts structure

Five invariant parameters of the stationary distributions of the ERB ions in the range $\mu = 0.01$–50
keV/nT at $L > 2$–3 were found as result of cross-analysis of satellite's data in (Kovtyukh, 1984,
1985a, 1985b, 1989, 1999a). For this analysis were used data of the satellites Explorer-12 (1961),
Explorer-14 (1962), Injun-4 (1965–1966), Explorer-33 (1966), Injun-5 (1968), ESRO-2 (1968),
1968-26B (1968), OVI-19 (1969), Molniya-1 (1970–1974), Explorer-45 (1971–1972), 1972-076B
(1972–1973), ATS-6 (1974), Molniya-2 (1975), ISEE-1 (1977–1979), SCATHA (1979),
AMPTE/CCE (1984-1986), Gorizont-21 (1985-1986), Akebono (1989-1991), CRRES (1991),
Gorizont-35 (1992) and ETS-VI (1994).
Invariant parameters of the ERB ion fluxes structure include the following quantities:
$\mu_m$ is corresponds to maximum at $E_m(L)$ in the energy spectra (exist only for solar origin ions);
$\mu_0$ is corresponds to index $E_0(L)$ of the exponential part of the energy spectra at $E_m < E < E_b$: $J \propto$
$\exp(-E/E_0)$;
$\mu_b$ is corresponds to the boundary $E_b$ between exponential spectral segment and a power-law tail;
$\gamma$ is corresponds to exponent of the power-law tail of the energy spectra: $J \propto E^{-\gamma}$;
$\xi_i$ is corresponds to the ratio of parameters $\mu_m$, $\mu_0$, or $\mu_b$ for different ion components (scaling
parameter).
Specific values of $\mu_m$ and $\mu_b$ depends on geomagnetic and solar activity; sometimes the segment
$\mu_m < \mu < \mu_b$ degenerates, but usually it is clearly expressed and well approximated by the exponent
function.
These parameters of the ions radiation belts are invariants relatively $L$ shells, and are displayed
not only in the energy spectra, but also in all other ion distributions by $E$, $L$ and $B/B_0$ for all main
ion components of the ERB. Here $B$ and $B_0$ are values of the magnetic field at the point of
measurement and in the equatorial plane on the same $L$ shell. For each of the main ion components
of the ERB it was established that spatial, energy and pitch-angle distributions of fluxes are
connected to each other with by means of the invariant parameters $\mu_m$, $\mu_0$, $\mu_b$, $\gamma$ and $\xi_i$ (Kovtyukh,
1984, 1994, 1999a, 2001). These parameters are completely determines the spatial-energy structure
of the ERB ion fluxes in a wide region of the space $\{\mu, L\}$.
For protons with equatorial pith-angles $\alpha_0 = 90 \pm 50^{\circ}$, these parameters have the following
values:
$$\mu_m = 0.55 \pm 0.10 \text{ keV/nT},$$
$$\mu_b = 1.16 \pm 0.29 \text{ keV/nT},$$
$$\mu_0 = 0.29 \pm 0.10 \text{ keV/nT, for } \mu_m < \mu < \mu_b,$$
$$\gamma = 4.25 \pm 0.75, \text{ for } \mu > \mu_b.$$





For helium ions and for CNO group ions with $\alpha_0 = 90 \pm 50^{\circ}$ these parameters have following
values:
$$\mu_m/\xi_i = 0.5 \pm 0.2 \text{ keV/nT},$$
$$\mu_b/\xi_i = 1.4 \pm 0.8 \text{ keV/nT},$$
$$\mu_0/\xi_i = 0.3 \pm 0.2 \text{ keV/nT, for } \mu_m/\xi_i < \mu/\xi_i < \mu_b/\xi_i,$$
$$\gamma = 4.7 \pm 2.2, \text{ for } \mu/\xi_i > \mu_b/\xi_i.$$
According to Kovtyukh (1985a, 1985b, 1989, 1999a), parameters $\mu_0$, $\mu_b$ and $\gamma$ applicable for ion
distributions only at $L > 3$, and parameters $\mu_m$ and $\xi_i$ applicable only at $L > 3.5$. Such restrictions
connected with the ionization losses of ions.
The scatter of the values of these parameters exceed considerably the statistical errors of the
measurements and it is connected with the averaging the experimental data of many satellites for
thirty years. These measurements obtained in different phases of the solar activity, in different
ranges of $E$, $\alpha_0$, $L$ and $B/B_0$, by different devices which have different resolution, different periods
of accumulation and averaging of the data.
From minimum to maximum of solar activity, $\xi_i$ changes from $M_i$ to $Q_i$ ($M_i$ and $Q_i$ are a mass
and charge of ions with respect to the corresponding values for protons), the exponential segment
of the spectra become softer ($\mu_0/\xi_i$ decreases in 1.5–2.0 times), and $\mu_b/\xi_i$ increases slightly
(Kovtyukh, 1999a).
Parameter $\gamma$ depends slightly on $B/B_0$, but $\mu_m$, $\mu_0$ and $\mu_b$ decreases with increasing $B/B_0$ at $L <$
145 6.6.
Usually, $\mu_b \approx \gamma\mu_0$ (the exponential and the power-law parts of the spectrum smoothly transfer one into
other) and $\mu_m \neq \mu_0$. Since for the Maxwellian distributions $\mu_m = \mu_0$ and parameter $\xi_i$ should be close to
unity, which does not correspond to the experimental results, an exponential part of the ion spectra, as
well as a power-law tail, are non-equilibrium.
The simplest explanation of the exponential segment and maximum in the spectra of the ERB
ions is that they reflect, in some degree, the quasi-Maxwellian distributions of the ions in the solar
wind, the main source of the ERB ions. The power-law tail of the ions ERB spectra formed, most
likely, by statistical mechanisms of particle acceleration in the magnetosphere (see Sect. 4 for more
details).

## 3 Modelling of the spatial-energy structure of ion fluxes in the radiation belts

As examples of modeling of the ERB for ions with $Z \geq 2$, in this section are presented the
distributions of the fluxes of helium ions and ions of the CNO group in the spaces of variables {$E$,
$L$} and {$L$, $B/B_0$}. Such presentations of fragmentary experimental data obtained in different ranges
of $E$, $L$ and $B/B_0$ are the most capacious envelopes of these data and make it possible to organize
them most effectively. For comparison, there are presented also the distributions of the fluxes of
protons of the ERB, which constructed by the same method. All of these distributions based on the
satellite data averaged for quiet periods.
For these distributions, only reliable data on ion fluxes were used which obtained in those
regions of $E$, $L$ and $B/B_0$ where these fluxes not distorted by the background of other particles. For
helium ions and for CNO group ions one have much smaller such data than for protons. In many
important experiments, the instruments did not allow separate fluxes of ions with $Z \geq 2$ by charge
of ions. For ions of the CNO group, separation by mass also were not performing usually. For
heavier ions, for example for Fe ions, we have even smaller such data. Therefore, this paper





presents only helium ions (without separating them by charge) and ions of CNO group (without
separating them by mass and charge).
Figures 1–6 shows the experimental results for differential fluxes of protons, helium ions, and
ions of the CNO group in the ERB near the plane of the geomagnetic equator, averaged for quiet
geomagnetic conditions and presented in space $\{E, L\}$. The values $E$ and $L$ presented in the
logarithmic scales. The ion fluxes $J$ have a dimension (cm$^2$ s ster МэВ/n)$^{-1}$ and corresponds to the
energies $E$ (MeV/n) and the equatorial pitch-angle $\alpha_0 \sim 90^{\circ}$. In some cases in these figures, an
average ion energies in the instrument channels corrected for the steepness of corresponding
energy spectra. For the data on these figures, lines of equal intensity of ion fluxes are plotted (by
the least squares method), and a decimal logarithms of the fluxes are shown near each line.
There can be trapped on the drift shell only ions with energy less than some maximum values
determining by the Alfvén's criterion: $\rho_i(L, E, M_i, Q_i) \ll R_c(L)$, where $\rho_i$ is the gyroradius of ions,
and $R_c$ is the radius of curvature of the magnetic field near the equatorial plane. According to this
criterion and the theory of stochastic motion of particles, a geomagnetic trap can capture and
durably hold only ions with $E$ (MeV) $< 2000 \times (Q_i^2/M_i) \, L^{-4}$ (Ilyin et al., 1984). This boundary, for
protons and atomic nuclei, presented in Figs. 1–6 by the green line.
Figures 1 and 2 shows exp results for proton fluxes in space $\{E, L\}$ averaged for quiet periods.
Figure 1 presents a results of the satellites 1968-81A (Stevens et al., 1970), Injun-5 (Krimigis,
1970; Venkatesan and Krimigis, 1971; Pizzella and Randall, 1971), OV1-19 (Croley et al., 1976),
Azur (Hovestadt et al., 1972; Westphalen and Spjeldvik, 1982), Molniya-1 (Panasyuk and
Sosnovets, 1973), GEOS-2 (Wilken et al., 1986), CRRES (Albert et al., 1998; Vacaresse et al.,
1999), GEO-3 (Selesnick et al., 2010) and Van Allen Probes (Selesnick et al., 2014, 2018),
obtained near maxima of solar activity in 20[th], 22[th], 23[th] and 24[th] solar cycles (1968–1971, 1990–
1991, 2000, 2012–2017).
Figure 2 presents a results of the satellites Relay-1 (Freden et al., 1965), OHZORA, ETS-VI and
Akebono (Goka et al., 1999), obtained near minima of solar activity between 19[th] and 20[th], 21[th] and
22[th], 22[th] and 23[th] solar cycles (1963, 1984–1985, 1994–1996).
The data of the satellites Explorer-45 (Fritz and Spjeldvik, 1979, 1981; Spjeldvik and Fritz,
1983) and ISEE-1 (Williams, 1981; Williams and Frank, 1984) refer to the years with intermediate
solar activity (1971–1972, 1977–1978) and at $L > 2.5$ are used in Figs. 1 and 2. The dependence of
proton fluxes on solar activity variations rapidly decreases with increasing $L$, and at $L > 2$ these
variations practically does not show in the satellite data (see, e.g., Vacaresse et al., 1999).
From the comparison of Figs. 1 and 2, one cand an see that at $L < 2.5$, and especially on $L < 1.4$,
the proton fluxes in the minima of solar-cyclic variations (Fig. 2) are higher than in the maxima of
solar activity (Fig. 1). Moreover, at the minima of solar activity, the inner edge of a proton flux
radial profiles with $E > 1$ MeV is less steep and achieves smaller $L$ shells.
Solar-cyclic (11-year) variations of proton fluxes with $E > 1$ MeV in the inner region of the
ERB connected mainly with the variations in the concentrations of atoms in the atmosphere (see,
e.g., Pizzella et al., 1962; Hess, 1962; Blanchard and Hess, 1964; Filz, 1967; Nakano and
Heckman, 1968; Vernov, 1969; Dragt, 1971; Huston et al., 1996; Vacaresse et al., 1999;
Kuznetsov et al., 2010; Qin et al., 2014). These variations achieves one order of magnitude at $L =$
1.14 and reduced rapidly with increasing $L$ (see, e.g., Vacaresse et al., 1999). The shape of the
proton energy spectra also undergo by solar-cyclic variations in the inner region of the ERB.
The atmospheric density depends on the intensity of the ultraviolet radiation of the Sun and
determines the loss rates of the ERB protons. Decreases in the amplitude of these variations with $L$
is connected with increases in the lifetime of protons with increasing $L$; at $L > 2$, this time
approach to the main period of the solar cycle. In these variations expressed some inertness of the
changes in the atmospheric density, and this lag increases with increasing $L$.
In Figs. 1 and 2, the red line shows the proton energy values corresponding to the average value
of the invariant $\mu_b$, and the blue line corresponds to the average value of the invariant $\mu_m$; these





219 values obtained from experimental data in (Kovtyukh, 1984, 1985a, 1985b, 1989, 1999a) and
220 presented in Sect. 2. These lines do not distinguished in Figs. 1 and 2, since the solar-cyclic
221 variations of the ion fluxes in the outer belt (at $L > 2.5$) are significantly weaker than in the inner
222 belt (see, e.g., Vacaresse et al., 1999). The maximum deviations from the mean values of $\mu_b$ and $\mu_m$
223 plotted on the red and blue lines with vertical segments; on a logarithmic scale, the magnitudes of
224 these segments do not depend on $L$ shell.

225 The isolines of the proton fluxes in Figs. 1 and 2, at $L > 3$ above the red line ($\mu > \mu_b$), go almost
226 parallel to it and are separated from each other approximately equal distances on a logarithmic
227 scale of energy. This conforms to adiabatic transformations of fluxes for the energies
228 corresponding to the power-law tail of the proton spectra. It results from these figures that in the
229 region of a near-dipole magnetic field, at $L = 3$–6, the parameter $\gamma = 4.8 \pm 0.5$. At $L > 6$, the
230 distance between these isolines increases with $L$, and the parameter $\gamma$ decreases, from $\sim 4.7$–5.0 at
231 $L = 6$ to $\sim 4.1$–4.5 at $L = 8$. This is due to the deviation of the magnetic field from the dipole
232 configuration ($L$ shells correspond here to the dipole magnetic field), as well as to the increasing
233 variability of this field with increasing $L$. In the interval between the red and blue lines ($\mu_m < \mu <$
234 $\mu_b$), the spectra have a close to exponential form and corresponds to $\mu_0 \sim 0.32$ keV/nT.

235 Thus, the values of the invariant parameters $\mu_b$, $\mu_0$ and $\gamma$ of the proton flux distributions in the
236 ERB, obtained from Figs. 1 and 2, are in good agreement with the values of these parameters given
237 in Sect. 2 and obtained in (Kovtyukh, 1985a, 1985b, 1989, 1999a) by other methods and with other
238 set of the experimental data.

239 However, a data representation, accepted here, does not allow determine the values of the
240 parameter $\mu_m$ for protons of the ERB and compare these with the values given in Sect. 2. For this,
241 it is necessary to reduce the step between the isolines of the fluxes ($\Delta \log J$) of protons with $E < 1$
242 MeV at $L > 3.5$ by $\sim 10$ times, which leads to significant systematic errors in such representation of
243 the data.

244 In addition, it must take into account that the ionization loss mechanisms generates in the proton
245 spectrum of the ERB also a less energetic maximum at $L < 5.5$ (this maximum considered in
246 details in Kovtyukh, 1989). Its energy increases from $\sim 0.01$–0.03 MeV at $L = 5.3$ to $\sim 0.4$ MeV at
247 $L = 3.2$ (the values of the adiabatic maximum on these $L$ are $\sim 0.11$ MeV and $\sim 0.52$ MeV,
248 respectively). These two maxima in the spectra of protons at $L \sim 3$–5 separated by a small local
249 minimum and, at insufficiently high resolution of the spectrometers, have a view an extended
250 plateau in the spectra.

251 According to the experimental data considered in (Kovtyukh, 1985a, 1985b, 1999a), it was found
252 that the parameters $\mu_b$ and $\gamma$ are detected only at $L > 3$. Here are considered more complete data of
253 satellites, and these parameters for protons with $E > 10$ MeV can be traced to $L \sim 2$: at $L = 2$, $\gamma = 4.7 \pm$
254 1.3 (Fig. 1) and $\gamma = 4.4 \pm 0.6$ (Fig. 2). This is due to the fact that, compare with (Kovtyukh, 1985a,
255 1985b, 1999a), the energy range here is significantly extended toward higher energies, but the
256 ionization losses rapidly decreases with increasing of the energy of the ERB protons (see, e.g., Schulz
257 and Lanzerotti, 1974; Kovtyukh, 2016a, 2016b).

258 Figure 2 show that at very high energies the proton spectra tail becomes steeper, which
259 corresponds to the limit of the magnetic confinement of protons in the ERB.

260 Figures 3 and 4 show averaged stationary fluxes of helium ions in the space $\{E, L\}$.

261 Figure 3 presented the data of the satellites OV1-19 (Blake et al., 1973; Fennell and Blake
262 1976), Explorer-45 (Fritz and Spjeldvik, 1978, 1979; Spjeldvik and Fritz, 1981) and SCATHA
263 (Blake and Fennell, 1981; Chenette et al., 1984), obtained near maxima of solar activity in 20[th] and
264 21[th] solar cycles (1968–1971, 1979).

265 Figure 4 presented the data of the satellites Molnija-2 (Panasyuk et al., 1977), Prognoz-5
266 (Lutsenko and Nikolaeva, 1978), ISEE-1 (Hovestadt et al., 1981) and Akebono (Goka et al., 1999),
267 obtained near minima of solar activity between 20[th] and 21[th] and between 22[th] and 23[th] solar cycles
268 (1975–1977, 1996).

From the comparison of Figs. 1–4, it can see that for helium ions with $E > 1$ MeV/n at $L \sim 2$–3
an amplitude of the solar-cyclic (11-year) variations is more than for protons. This difference
connected with the ionization losses of the ERB ions: for helium ions these losses more than for
protons.
In Figs. 3 and 4, the red line shows an energy of the helium ions corresponding to the average
value of the invariant $\mu_b/\xi_i$, and the blue line corresponds to the average value of the invariant
$\mu_m/\xi_i$, which are represent in Sect. 2. Here $\xi_i = Q_i$ for Fig. 3 and $\xi_i = M_i$ for Fig. 4. It was taken into
account that for $E > 0.2$ MeV/n at $L < 6$ the average (main) charge of helium ions is $Q_i = +2$ (see,
e.g., Spjeldvik, 1979). The maximum deviations from the mean values of $\mu_b/\xi_i$ and $\mu_m/\xi_i$ at energy
scale plotted on the red and blue lines with vertical segments; on a logarithmic energy scale, the
magnitudes of these segments do not depends on $L$ shell. The isolines of the helium ions fluxes in
Figs. 3 and 4 at $L > 2$ pass above the red line almost parallel to it, and average value of the
parameter $\gamma \sim 5.5$.
Thus, the values of the invariant parameters of the power-law tail of the spatial-energy flux
distributions of helium ions in the ERB, obtained from Figs. 3 and 4 are in good agreement with
the values of these parameters given in Sect. 2 and obtained in (Kovtyukh, 1999a) by other
methods and with other compositions of experimental data. To find the values of the parameters
$\mu_m/\xi_i$ and $\mu_0/\xi_i$ from Figs. 3 and 4, it is necessary to reduce the step between the isolines of the ion
fluxes significantly, which leads to large systematic errors in such representation of the data.
Figures 5 and 6 show the results for the average stationary fluxes of ions of the CNO group in
the space $\{E, L\}$.
Figure 5 presented the data of the satellite Explorer-45 (Spjeldvik and Fritz, 1978; Fritz and
Spjeldvik, 1981), obtained near maximum of solar activity in 20[th] solar cycle (1971–1972).
Figure 6 presented the data of the satellites ATS-6 (Spjeldvik and Fritz, 1978; Fritz and
Spjeldvik, 1981) and ISEE-1 (Hovestadt et al., 1978), obtained near the minimum of the solar
activity between 20[th] and 21[th] solar cycles (1974–1975, 1977).
In Figs. 5 and 6, the red line shows the CNO ions energy values corresponding to the average
value of the invariant $\mu_b/\xi_i$, and the blue line corresponds to the average value of the invariant
$\mu_m/\xi_i$, which presented in Sect. 2. Here $\xi_i = Q_i$ for Fig. 5 and $\xi_i = M_i$ for Fig. 6. It was taken into
account that for $E > 0.1$ MeV/n at $L \sim 3$–5 the average charge of the CNO group ions is $Q_i = +4$
(see, e.g., Spjeldvik and Fritz, 1978). The maximum deviations from the mean values of $\mu_b/\xi_i$ and
$\mu_m/\xi_i$ at energy scale plotted on the red and blue lines with vertical segments; on a logarithmic
energy scale, the magnitudes of these segments do not depends on $L$ shell. The isolines of the CNO
ions fluxes in Figs. 5 and 6 at $L > 3$ pass almost parallel to each other.
However, at the maximum of solar activity their slope on $L > 3$ is significantly less than the
slope of the red line, which indicates more significant ionization losses of ions of the CNO group
at $L = 3$–5 compared to these losses for protons and for helium ions. According to these results, at
$L \sim 3$–6 for ions of the CNO group, the average value of the parameter $\gamma \sim 6$.
Thus, the values of the invariant parameters of the power-law tail of the spatial-energy flux
distributions of the CNO group ions in the ERB, obtained from Figs. 5 and 6, are in good
agreement with the values of these parameters given in Sect. 2 and obtained in (Kovtyukh, 1999a)
by other methods and with other set of experimental data. To find the values of the parameters
$\mu_m/\xi_i$ and $\mu_0/\xi_i$ from Figs. 5 and 6, it is necessary to reduce the step between the isolines of the ion
fluxes significantly, which leads to large systematic errors in such representation of the data.
As well as for protons (Figs. 1 and 2), for helium ions (Figs. 3 and 4) and CNO group ions
(Figs. 5 and 6), with $E > 0.1$ MeV/n at small $L$, fluxes during solar minimum more, than during
solar maximum. The larger the atomic number $Z$ of the ERB ions, the greater the amplitude of
these variations. The solar-cyclic variations of ion fluxes explained by the same mechanism of
ionization losses of particles, which proposed for the ERB protons (see above). The main loss





mechanism of ions at $E > 0.1$ MeV/n at $L < 3.5$ are Coulomb losses, for which the loss rate
increases rapidly with increasing $Z$ of the ions (as $Z^2$).
Figs. 7–9 show the experimental results for differential fluxes of protons and helium ions in the
ERB, averaged for quiet geomagnetic conditions and presented in the space $\{L, B/B_0\}$ for different
ion energies (as examples). The values of $L$ and $B/B_0$ are plotted on logarithmic scales. For these
results, the lines of equal intensity of ion fluxes are made, and decimal logarithms of the fluxes are
shown near each of these line.
Figure 7 presents stationary fluxes of protons with $E = 0.4$ MeV, averaged for quiet periods by
the data of the satellites Injun-5 (Krimigis, 1970; Venkatesan and Krimigis, 1971; Pizzella and
Randall, 1971), Molniya-1 (Panasyuk and Sosnovets, 1973) and GEOS-2 (Wilken et al., 1986).
These data obtained near the maxima of solar activity in 20th and 21th solar cycles (1968–1970,
329 1978).

Figure 8 presents stationary fluxes of protons with $E = 0.4$ MeV, averaged for quiet periods by
the data of the satellites 1964-45A (Mihalov and White, 1966), ISEE-1 (Williams, 1981; Williams
and Frank, 1984) and Polar (Walt et al., 2001). These data obtained near the minima of the solar
activity between 19th and 20th, between 20th and 21th and between 22th and 23th solar cycles (1964,
334 1977, 1998).

From Figs. 7 and 8 it can be seen, that for protons with $E = 0.4$ MeV during solar activity
minima fluxes are greater than during solar activity maxima, at the same points in the space $\{L,$
$B/B_0\}$; most significantly this discrepancy in proton fluxes is observed at $B/B_0 > 100$ (this is also
valid for other proton energies). This effect connected with the solar-cyclic variations of the
Earth's upper atmosphere temperature, as for protons of the ERB near the equatorial plane.
Figure 9 presents stationary fluxes of helium ions with $E = 0.2$ MeV/n, averaged for quiet
periods by the data of satellite Polar (Spjeldvik et al., 1999) which were obtained near the
minimum of the solar activity between 22th and 23th solar cycles (1996). For equatorial plane ($B/B_0$
= 1) we used data of satellite Explorer-45 presented in (Fritz and Spjeldvik, 1978, 1979; Spjeldvik
and Fritz, 1981).
From Figs. 7–9 it seen that with increasing $B/B_0$ the maximum of the ion fluxes shifts to a
higher $L$. This performed also for other ion energies in the range of 0.1–100 MeV/n and connected
with increasing in ionization losses and decreasing in the radial diffusion rate of the ERB ions with
increasing $B/B_0$.
With decreasing altitude of the observation point (at a given $L$), the concentration of atoms and
ionization losses increases, which lead to the formation an altitude dependence of the ion fluxes:
with decreasing altitude (at fixed $L$ and $E$), ion fluxes of the ERB are decreases. With decreasing $L$,
this dependence enhances and, consequently, at low altitudes (at $h \sim 500$–$1000$ km) a maximum of
the ERB should be located at larger $L$ compared to the equatorial plane (see Figs. 7–9).
When the exosphere is heated, the altitude dependence of a concentration of atoms becomes
weaker. Therefore, the difference in the position of the ERB maximum in the equatorial plane and
at low altitudes decreases (see Figs. 7 and 8).
Since reliable experimental data on helium ions in the ERB are insufficient, the distributions of
their fluxes in space $\{L, B/B_0\}$, especially at higher energies, have more or less significant lacunae.
The most complete distributions of these ions in the space $\{L, B/B_0\}$ obtained only for $E < 1$
MeV/n in the minimum of solar activity. On ions of the CNO group one have even less reliable
and complete data. The distributions of these ions in the space $\{L, B/B_0\}$ represent a very
variegated picture. However, it is possible to conclude from these incomplete data that for higher
helium ion energies, as well as for ions of the CNO group, the pattern of flux distribution in space
$\{L, B/B_0\}$ is similar in form to that shown in Figs. 7–9.
Patterns of isolines of fluxes of the ERB ions in Figs. 1–9 are almost identical in shape for
various ionic components, especially for more complete data of protons and helium ions on $L > 2$,
and with increasing $E$ and $L$ the degree of such similarity increases. Some deviations of these



isolines from the overall picture for ions with $Z > 2$ are mainly due to the deficiency of
experimental data. This similarity has a basis in the unity of the main source (solar wind) and on
the unity mechanisms of transfer, acceleration and losses of ERB ions (radial diffusion, betatron
acceleration and ionization losses).
The absence of ions with $Z \geq 2$ at $L < 2$ (or very low values of these fluxes) is explained the fact
that their lifetimes, determined by ionization losses, are much less than the same times for protons.
Besides, for protons at $L < 2$ there is an additional source, CRAND (for more on this, see Kovtyukh,
Kovtyukh, 2018).

## 4 Discussion

We consider here the main consequences of these results for the physics of the magnetosphere.
Since parameters $\mu_b$ and $\gamma$ of the power-law tail of the ion spectra are invariant with respect to
$L$, it can be assumed that this part of the ERB ion spectra formed in the plasma sheet (PS) in the
tail of the magnetosphere which is adjacent to the geomagnetic trap. High-energy part of the ion
spectra in the PS, at $R \sim 20$–$40$ $R_E$, has approximately the same shape as in the ERB and the
average values of the parameters $\mu_b$ and $\gamma$ are close to our estimates of these parameters for the
ERB. Moreover, the proton spectra in the ERB are consistent with the PS spectra not only in form
but also in absolute fluxes: they can be obtain from the PS ion spectra in result of the simplest
adiabatic transformations (for more details see in Kovtyukh, 1999b, 2001).
According to the data of the satellites IMP-7, IMP-8 (Sarris et al., 1981; Lui et al., 1981) and
also ISEE-1 (Christon et al., 1991) for ion spectra of the PS, the typical substorms not changed of
the spectral shape and cause only parallel shift of the spectra along logarithmic axes $E$ and $J$ (for
ions with $\mu/\xi_i > 0.5$ keV/nT). These results are point out that the time scales of the processes of
formation of the power-law tail of the ion spectra in the PS far exceed the characteristic times of
substorms.
Invariant parameters $\gamma$ and $\mu_b$ of the power-law tail of the ion spectra reflect, apparently, the
most fundamental features of the mechanisms of acceleration of ions in the tail of the
magnetosphere. One can try to connect the obtained values of these parameters with the most
general presentations on the mechanisms and character of ion acceleration in the PS of the
magnetospheric tail.
Most likely, the tail of ion energy spectra of the PS formed by statistical mechanisms of the ion
acceleration. This supported by many experimental results.
Statistical character of these mechanisms reveal itself, in particular, in the fact that the ratios of
fluxes (and partial concentrations) of ions with different $Z$ at low and high energies can differ
greatly. During wandering in phase space, ions gradually forget their origin, and, therefore, the
high-energy tails of the ion spectra do not contain unambiguous information on the partial
concentrations of different components of ions in there source (for more details, see Kovtyukh,
1999b, 2001).
Most likely, the high-energy part of the ion spectra of the PS formed by the mechanisms of
acceleration of particles on magnetic irregularities moving relative to each other (Fermi
mechanism). If mass of the ions are small compare to masses of the magnetic irregularities in the
PS, the average values of the exponent $\gamma$ of the power-law tail of the spectra should not depend on
mass and charge of these ions.
Under equilibrium conditions, this parameter determined by the average fraction $\bar{\beta}$ of energetic
ions in the total energy density of particles and magnetic irregularities. From the theory developed
on these fundament by Ginzburg and Syrovatskii (1964), it follows: $\gamma - 1 \approx (1 - \bar{\beta})^{-1}$. With
increasing $\bar{\beta}$ in the interval $0 < \bar{\beta} < 1$, the value $\gamma$ monotonically increases and $\gamma \to \infty$ for $\bar{\beta} \to$
1. For real average values $\bar{\beta}$ in the central PS, we get $\gamma \sim 3.5$–$7.0$ ($\gamma \sim 4.3$ at $\bar{\beta} \sim 0.7$).





Compared with the power-law tail, a short quasi-exponential segment of the ion spectra allows
several different interpretations. Remaining within the framework of the most general physical
concepts about the mechanisms of acceleration of cosmic plasma particles, the presence of this
segment in the ion spectra of the PS and ERB one can explain in the terms of the quasi-particles
theory. The structure of the magnetic field of the PS one can represent as quasi-periodic spatial
grilles with different periods nested into each other (fractality) and as an energy of ions increases,
grilles with more and more long periods gradually fall out from the process of acceleration of ions.
Then the upper boundary of the exponential segment of the spectra corresponds to ions for which
all small-scale grilles are transparent: ions with $\mu > \mu_b$ detect only the most large-scale grille. The
fractal topology of the PS on scales from $\sim 0.4$ to $\sim 8$ thousand km reveal itself, for example, in the
results of the satellite Geotail (Milovanov et al., 1996).
Spectra with a power-law tail and a quasi-exponential segment at lower energies generate when
a value $\Delta B / \bar{B}$ for magnetic irregularities is proportional to their size $\delta r$ and at $\delta r < r_s$ the spectral
density of irregularities rapidly decreases with increasing $\delta r$, and at $\delta r > r_s$ remains almost
unchanged. Apparently, the spectrum of magnetic irregularities in PS with thickness $r_s$ has just
such form.
Then the lower boundary $\mu_b$ of the power-law tail corresponds to the condition $r_s/\rho_i \sim 10$ ($\rho_i$ is
the gyroradius of ions), i.e. $\mu_b \sim 0.02(Q_i^2/M_i)B_s r_s^2$ keV/nT, where $B_s$ is the average magnetic field
induction in the PS (in nT) and $r_s$ is normalized to the Earth's radius. Believing $B_s \sim 30$ nT and $r_s \sim$
1.3 $R_E$, one obtain $\mu_b \sim 1.0$ $(Q_i^2/M_i)$ keV/nT (for details, see Kovtyukh, 1999b).
Thus, the main invariant parameters of the ERB structure one can relate with the average
physical properties of the PS.
Parameter $\gamma$ one can relate with the fraction of energetic ions in the total energy density of
particles and of a magnetic irregularities in the PS (Kovtyukh, 1999b, 2001).
Parameter $\mu_b$ one can relate with a thickness of the PS and a magnitude of its magnetic field
(Kovtyukh, 1999b, 2001).
Parameter $\mu_0$ one can relate with a small-scale structure (spectrum of turbulence) of the PS
(Kovtyukh, 1999b, 2001).
Parameter $\mu_m$ is well corresponded to maximum in the PS ion spectra and in solar wind
(Kovtyukh, 1989, 2001).

## 5 Conclusions

In this work, it was found that results of the measurements of the stationary fluxes of the main ion
components of the ERB (protons, helium ions and ions of the CNO group) line up in the certain
regular patterns in the spaces $\{E, L\}$ and $\{L, B/B_0\}$. It is reveal that such patterns is associated with
the existence of invariant parameters of the spatial-energy distributions of the ERB ion fluxes and
the values of these parameters are determined.
Earlier, the results of systematization of the spatial-energy distributions of the main ionic
components of the ERB and finding of their invariant parameters were presents in (Kovtyukh,
1985a, 1985b, 1999a, 2001). Here, the experimental database is significantly expanded, many
modern measurements of the ion fluxes of the ERB have been added, and all results are presented
in a more general and visual form.
Solar-cyclic (11-year) variations of the spatial-energy distributions of the ERB ion fluxes and
their invariant parameters are considered. It is shown that the larger an atomic number $Z$ of the
ERB ions, the greater the amplitude of these variations. This is also typical for faster variations in
the fluxes of the ERB ions, during geomagnetic storms and other disturbances of the Earth's
magnetosphere, which is underlined in the review (Kovtyukh, 2018).





The results presented here show that when constructing realistic multicomponent models of the ERB ion fluxes based on limited and incomplete experimental data, the invariant parameters of the ion distributions of the ERB can serve as a pattern of these fluxes distributions.

Our drawings have also revealed the localization of "white spots", especially extensive for ions with $Z \geq 6$, which should be filled on the results of the future experiments on the satellites.

The physical mechanisms leading to the formation of the invariant structure of the ERB are considered. It is shown that energy spectra of the ERB ions with $\mu/\xi_i > 0.5$ keV/nT can be generated in the plasma sheet (PS) of the tail of the Earth's magnetosphere. In the geomagnetic trap these spectra transforms adiabatically, and ions are loss part of their energy by ionization and other loss mechanisms.

**Acknowledgements** This work was supported by Russian Foundation for Basic Research RFFI grant No. 17-29-01022.

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

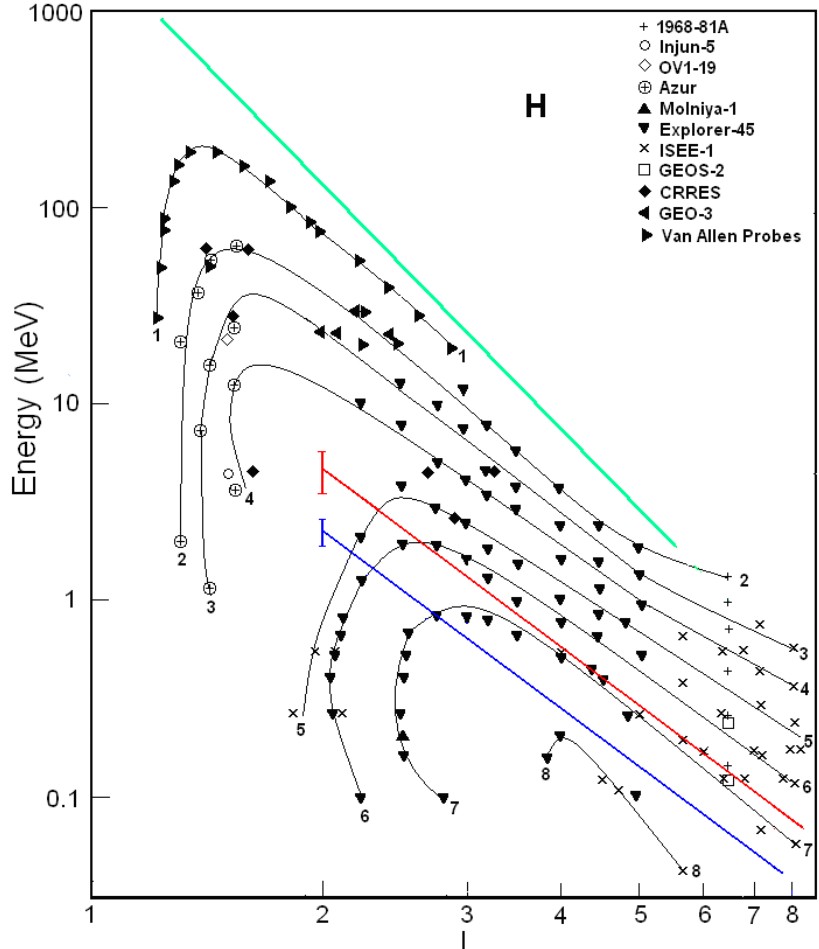

**Figure 1.** Proton fluxes in the ERB near maxima of a solar activity. A numbers on the curves are equal the values of a
decimal logarithms of $J$ where $J$ in (cm$^2$ s ster МэВ)$^{-1}$ are differential fluxes of protons with $\alpha_0 \approx 90^\circ$. The data of
different satellites presented by different symbols.

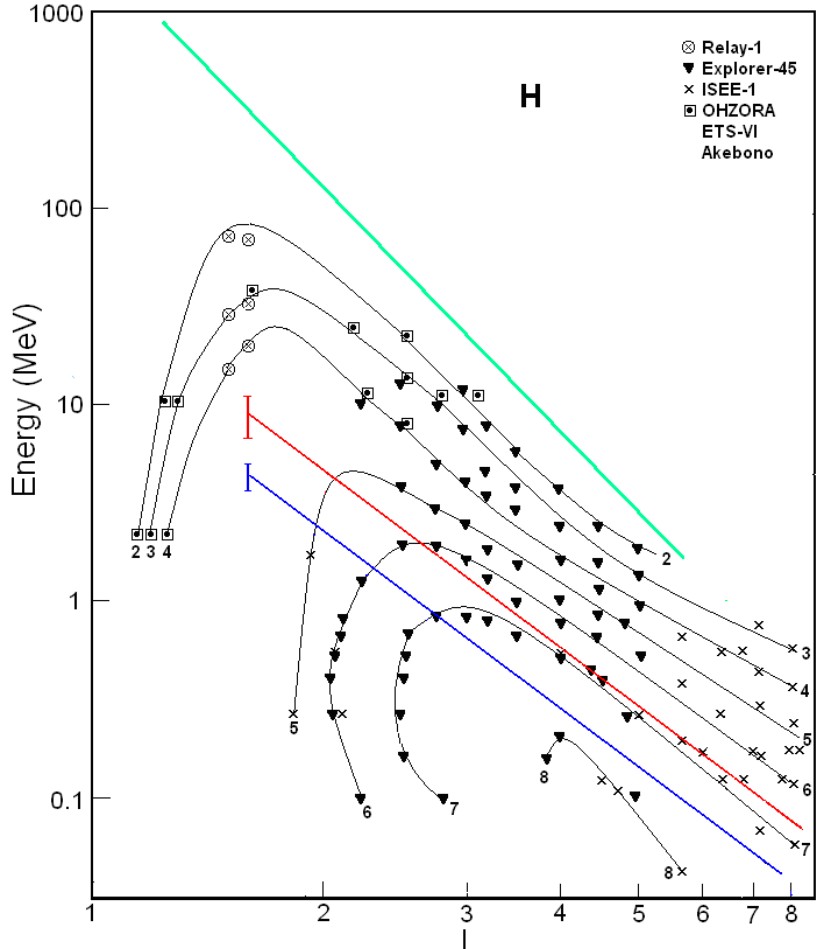

**Figure 2.** Proton fluxes in the ERB near minima of a solar activity. A numbers on the curves are equal the values of a
decimal logarithms of $J$ where $J$ in $(cm^2 \ s \ ster \ MэB)^{-1}$ are differential fluxes of protons with $\alpha_0 \approx 90^{o}$. The data of
different satellites presented by different symbols.

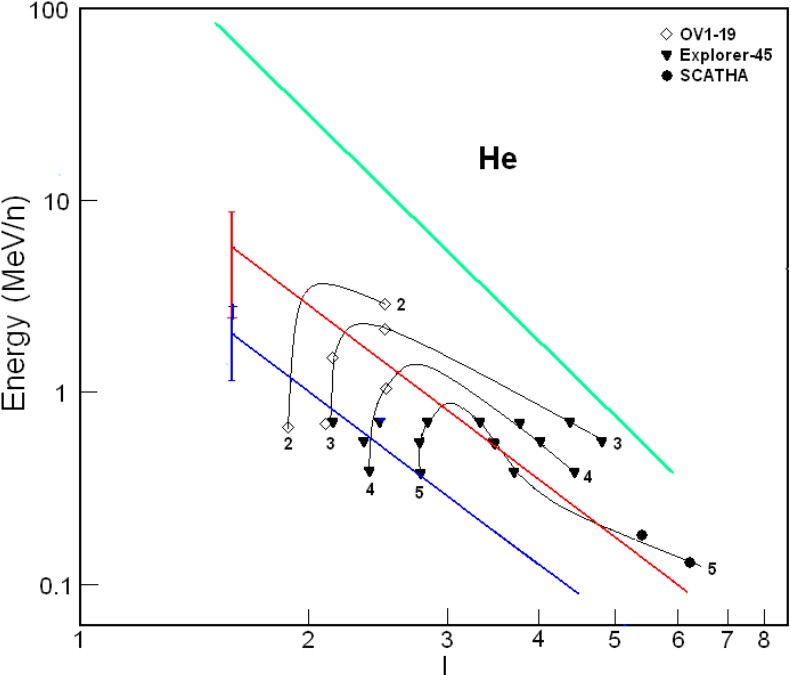

**Figure 3.** Helium ion fluxes in the ERB near the maxima of a solar activity. A numbers on the curves are equal the value of a decimal logarithms of $J$ where $J$ in (cm$^2$ s ster MэB/n)$^{-1}$ are differential flux of protons with $\alpha_0 \approx 90^\circ$ (near the plane of the geomagnetic equator). The data of different satellites presented by different symbols.

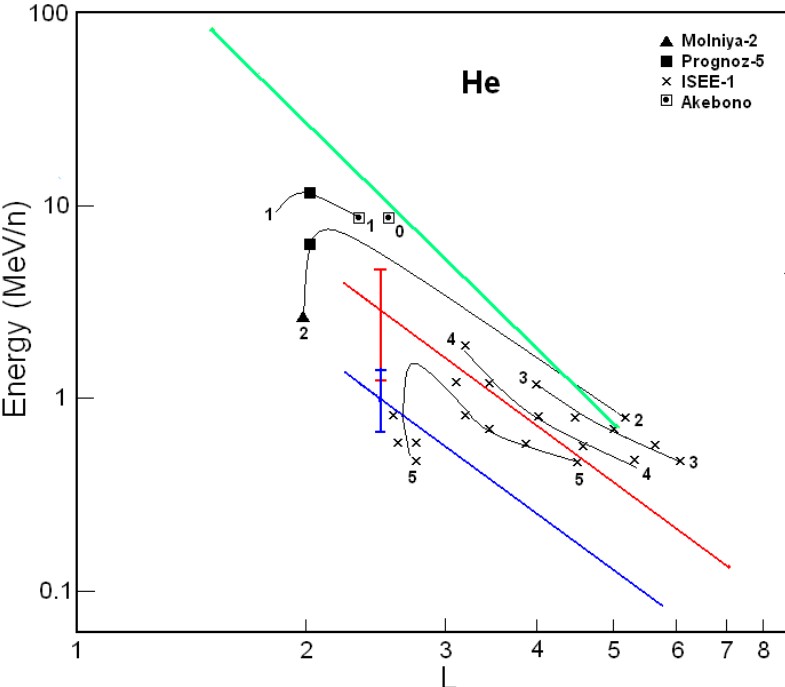

**Figure 4.** Helium ion fluxes in the ERB near the minima of a solar activity. A numbers on the curves are equal the values of a decimal logarithms of $J$ where $J$ in (cm$^2$ s ster MэB/n)$^{-1}$ are differential flux of protons with $\alpha_0 \approx 90^\circ$ (near the plane of the geomagnetic equator).The data of different satellites presented by different symbols.



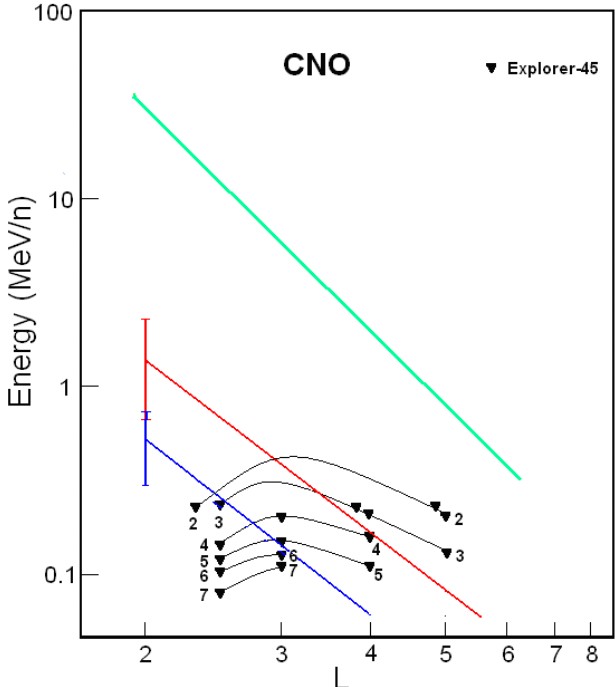

**Figure 5.** CNO ion fluxes in the ERB near the maximum of a solar activity. A numbers on the curves are equal the
values of a decimal logarithms of $J$ where $J$ in $(cm^2 \text{ s ster МэВ/n})^{-1}$ are differential flux of protons with $\alpha_0 \approx 90^o$ (near
the plane of the geomagnetic equator). The data of different satellites presented by different symbols.

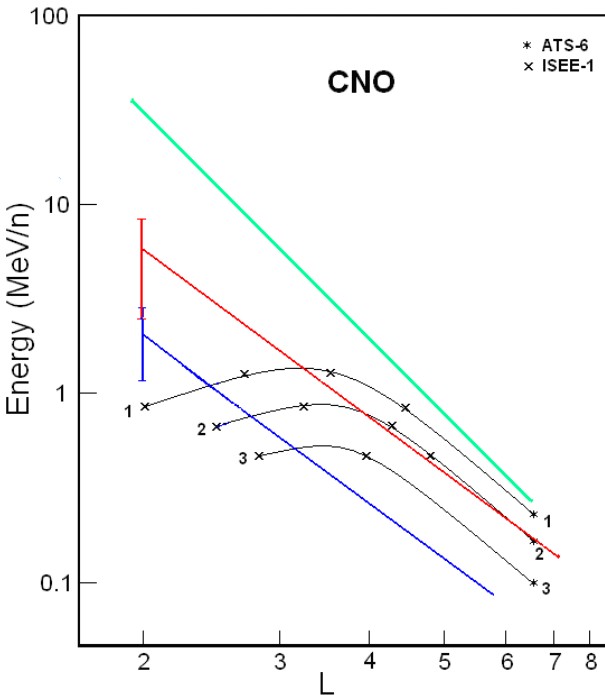

**Figure 6.** CNO ion fluxes in the ERB near the minimum of a solar activity. A numbers on the curves are equal the
values of a decimal logarithms of $J$ where $J$ in $(cm^2 \text{ s ster МэВ/n})^{-1}$ are differential flux of protons with $\alpha_0 \approx 90^o$ (near
the plane of the geomagnetic equator). The data of different satellites presented by different symbols.



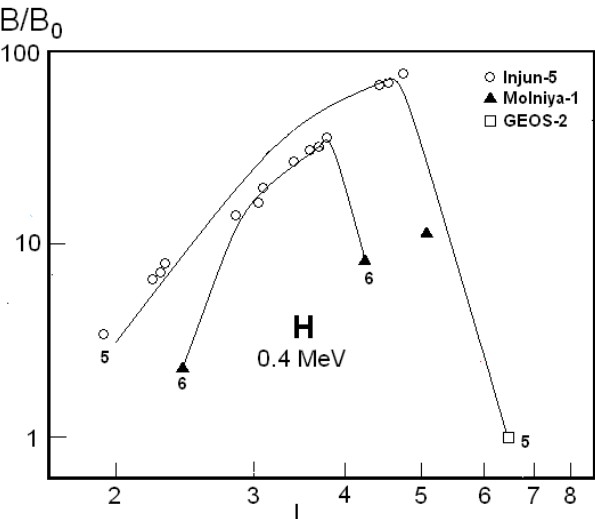

**Figure 7.** Average stationary fluxes of protons with $E = 0.4$ MeV in space $\{L, B/B_0\}$ near the maxima of solar activity.
A numbers on the curves are equal the values of a decimal logarithms of $J$ where $J$ in $(cm^2$ s ster MэB/n$)^{-1}$. The data of
different satellites presented by different symbols.

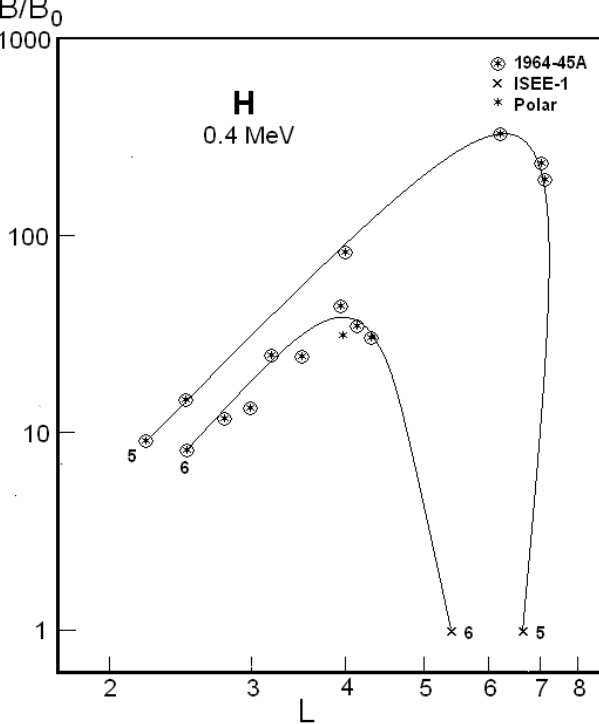

**Figure 8.** Average stationary fluxes of protons with $E = 0.4$ MeV in space $\{L, B/B_0\}$ near the minima of solar activity.
A numbers on the curves are equal the values of a decimal logarithms of $J$ where $J$ in $(cm^2$ s ster MэB/n$)^{-1}$. The data of
different satellites presented by different symbols.



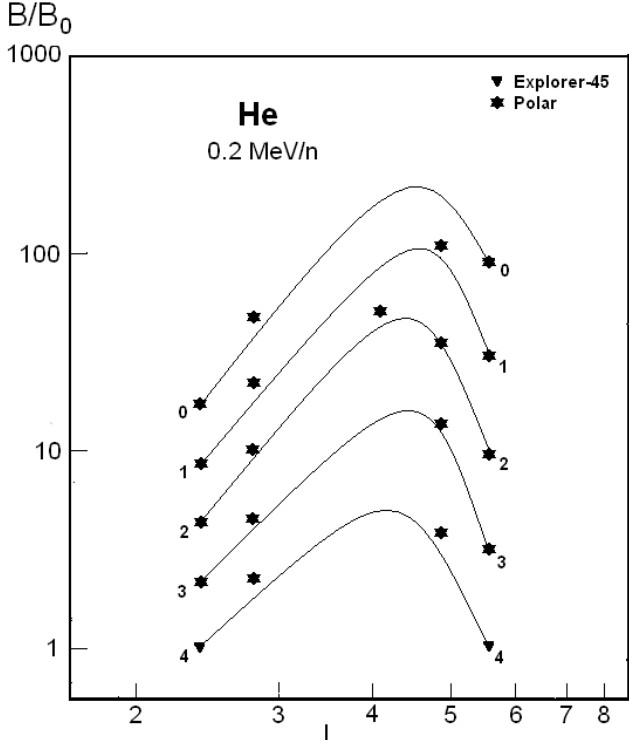

**Figure 9.** Average stationary fluxes of helium ions with $E = 0.2$ MeV/n in space $\{L, B/B_0\}$ near the minimum of a solar activity. A numbers on the curves are equal the values of a decimal logarithms of $J$ where $J$ in (cm$^2$ s ster МэВ/n)$^{-1}$. The data of different satellites presented by different symbols.