# Peer review of "Invariants of the Spatial-Energy Structure and Modeling of the Earth's Ion Radiation Belts"

_Annales Geophysicae, 2019_

## Referee Comment (RC1) · Anonymous Referee #1 · 4 Jul 2019

**General comments**

This paper presents a parametric model for the density of various ion species in the Earth radiation belts. This model describes the global structure of the radiation belts for protons, helium, and for ions of the CNO group. Based on extensive satellite data, the parameters of the models have been fitted, independently for the proton populations and for the other ions. The validity of the model is discussed, species by species, by comparison with in-flight data. The solar cycle dependency is presented. Finally, a physical interpretation of the model is detailed.

Overall, the paper is difficult to understand. Firstly, the English is very poor (see tech-

nical corrections hereafter), and difficult to grasp. Secondly, the model is not appropriately explained, as the author refers to previously published literature, which could not be accessed by this reviewer. Thirdly, as described in the specific comments, the figures are not very clear and supportive of the arguments developed in this paper.

**Specific comments**

Section 2 of this paper presents the model parameters and their measured values, but not the model itself, which is only suggested by the description of the parameters. A detailed and self-sufficient description of the model should be given in this section.

On section 3, numerous similar figures are presented. It is not clear how these figures were obtained from the data. In particular, these figures present iso-lines on power of tens, with most satellite data points placed on the iso-lines, which suggest some interpolation was done on the satellite data. Section 3 should detail how this figures were made.

The conclusions of the comparison of figures 1 to 4 proposed at line 269 are not clearly apparent in the figures. Similarly, the low-altitude effects described at line 352 cannot be clearly seen on the figures, because the transformation B/Beq to altitude is not straightforward (for instance, the 1000 km altitude line could be drawn on figure 7-9 to support the arguments of this paragraph).

A reference should be provided at line 347 for the dependency of the radial diffusion rate on B/Beq.

A figure supporting the information at line 362-364 about the CNO group data could be provided.

**Technical corrections**

Numerous English errors have been found, for instance on lines 21, 24, 34, 48, 63, 67, 69, 79, 118, 132, 134, 137, 161, 163, 166, 173, 185, 193, 201, 215, 220, 226, 239, 271, 290, 314, 338, 379, 387, 389, 398, 403, 469.

In the figure legends, the MeV unit is displayed in Cyrillic. Moreover, the model lines on the figures are not described in the legends.
* * *

---

## Author Comment (AC1) · 6 Jul 2019

I am very grateful to the Referee for these comments: they help clarify the essence of my work and I, of course, will make all the necessary changes in the revised manuscript. In many places of this manuscript, starting with the title, the word "model" is found and in almost all cases this word has the most general meaning here: any observation or measurement is a physical model (modeled by instruments or a brain). But in fact, no empirical or mathematical models are used in this manuscript, and what is presented in Fig. 1-9, these are collections of experimental data. Associations with models (in the narrow sense) also invoke the word "parameters". What presented

here, in section 2, and called the invariant parameters of ERB are not the parameters of mathematical or empirical models of ERB. These parameters obtained directly from the results of various experiments, from the figures of the corresponding articles. They were found for each energy spectrum and for each dependence from L of the ions fluxes, and only then these parameters were averaged (separately for each ion component). These invariants exist only in the region where the transport (radial diffusion) of ions dominates their losses. With increasing B/B0, the rate of radial diffusion of ions decreases, and the rate of their loss increases rapidly. Therefore, on small L and for large B/B0, these invariants are not applicable to ERB and in Figs. 7-9 they are not presented (these figures are given only for completeness). Author's papers in journals Cosmic Res. and Geomagn. Aeron. (in English) can be found in any major university library. But in order to make sure that there are such invariants and in the correctness of the values given here, it is not necessary to read these articles. It is enough to open several articles where there are experimental spectra or radial profiles of ion fluxes with E > 0.1 MeV for L > 3, obtained in quiet periods near the equatorial plane. I will take into account all these remarks and the words "model" and "parameter" will be saved only where it says about specific ERB models. I checked the descriptions of the figures and added them with the necessary explanations. I agree that I don't speak English well enough and gratefully accept any corrections to the text of the manuscript.

---

## Author Comment (AC2) · 6 Jul 2019

Specific comments

The invariants presented in Section 2 are not tied to any particular model (see above), but they can be used in works on the creation of both empirical and mathematical models of spatial-energy distributions of ERB ions. In particular, they were tested in many my works on modeling the pitch-angle distributions of protons, the empirical model of ion fluxes for region of the geosynchronous orbit (GSO), the daily course of ion fluxes on the GSO and in other problems. Points on fig. 1-9 were obtained from experimental radial profiles of differential ion fluxes. It was taken into account

that, for different authors, these fluxes have different dimensions. For example, for ions with Z > 1, these fluxes are given in (cm2 s ster MeV/n)–1 or in (cm2 s ster MeV)–1; in the latter case the same ion flux will have a value of Z times less. Iso-lines in Fig. 1-9 are envelopes of experimental points (they are constructed by the method of the ïАč2). It was important to choose a form of representation (space of variables) in which the results of different experiments (with different sets of the energy chennels) one could accommodate organically and without resorting to interpolations and extrapolations of the data. As such representation was chosen the space {E. L}. All other representations, such as radial profiles of ion fluxes for different energies, suggest interpolation and extrapolation of an experimental data; this is greatly complicates the solution of our problem and introduces large systematic errors and arbitrariness in the choice of curves approximation. To present the data obtained outside the equatorial plane, it was natural to use the space {B/B0, L} for ions of different energies. Here I had to resort to interpolation of data and Figs. 7–9 are less complete and accurate compared to Figs. 1–6. For comparison Figs. 1-2 with Figs. 3-4, specific values of proton and helium ion fluxes are given (lines 269-271). For Figs. 7-9 interpretation can be simplified, and then it is not necessary to put on these figures the dependences B/B0(L) for fixed heights. For the dependency of the radial diffusion rate on B/B0 a reference are added (at line 347). There are not enough data for CNO group ions and they are very fragmentary; build or them figures like to Fig. 7-9 while is impossible. One can only make assumptions on the basis of available data and general considerations. I changed the sentence at lines 362-364, made it more careful.

Technical corrections

I would correct the errors in the revised version of the manuscript. All dimensions of the MeV unit displayed in Cyrillic replaced by on Latino. In the caption to Figs. 1-6 added explanations to color lines. In Figs. 1–6, the color lines are show dependencies on L of the ion energies corresponding to the structure invariants (E ïĆţ ïĄ■L-3) and also the maximum energy of ions which can be trapped in the ERB(E ïĆţ L-4).

---

## Referee Comment (RC2) · Anonymous Referee #2 · 7 Jul 2019

**General comment**

The review written by the author and published last year in the journal Space Science Reviews (SSR) concluded that there is a need for the development of radiation belt heavy ion empirical models. The submitted paper gathers available heavy ion measurements obtained all over the space age and tries to pave the way for the development of such models. To do so, invariant parameters (that are constant over a given range of L) are detailed and available measurements are shown on Figures and discussed in detail. The solar cycle variation of heavy ion fluxes is for the first time explored in the submitted article. Finally, the presented measurement database is important to explore the physical mechanisms that govern the heavy ion radiation belts, what is done by the author in sections 3 and 4. Comparative lessons with what is known for the protons are drawn.

There is no doubt that the work presented here is important and may contribute to advances in our understanding and prediction of the heavy ion radiation belts. The submitted article may therefore, according to the reviewer, be ultimately published after several clarifications. The article needs to first be edited for English, in order to make it easily understandable so that it would have an impact on the work of others. The reviewer is then wondering: what is new in this article, compared in particular with the review published in SSR last year (see specific comments)? I recommend the article to be revised and reviewed again to see if, after English proof and clarifications, the article would be suitable for publication in Annales Geophysicae.

**Specific comments**

Section 2: would it be possible to clarify if the invariant parameter values given here come from previous publications or are the outputs of the new study? The values reported here are very important. If the values have now been recomputed or updated, would it be possible to have them highlighted in a table, for instance? A possible use from other researchers would be to compare them with what has been observed by now two orbiters around Jupiter, as these orbiters performed numerous observations of trapped heavy ions (helium, oxygen, sulfur).

Would it be possible to remind, with maybe one sentence, the criterion used to select quiet periods over which the measurements are averaged?

The measurements are averaged near solar cycle minimum and near solar cycle maximum. Are the measurements very dispersed around this average in each case or are the standard deviations small compared to the shown averages? A comment may be added in the main text about this.

The dataset of heavy ion measurements is limited, but is it large enough to conclude if there is any observable Magnetic Local Time asymmetry in the heavy ion radiation belts, in particular at the lowest considered kinetic energies?

Section 4: What are the new conclusions on the physics of the heavy ion radiation belts? If you confirm what has already been reported in previous publications, would it be possible to add a sentence to state it? Otherwise, new findings may be more highlighted in this section and in the conclusion.

Lines 453-454: "Here, the experimental database is significantly expanded, many modern measurements of the ion fluxes of the ERB have been added", what are the modern heavy ion measurements added since the article published by the author in 2001? For the protons, one can see the GEO-3 and Van Allen Probe observations, however there does not seem to be any "modern" measurement of heavier ions.

**Technical corrections**

The article needs to be edited for English.

It would help the reading to explain in the figure captions what the colored lines refer to, even if it is explained in the main text. In the main text, would it be possible to clarify what the maximum deviations shown by the colored vertical segments are: are they based on energy spectra measured by all the satellites, or only a subset? Would it also be possible to clarify the meaning of the following statement "on a logarithmic energy scale, the magnitudes of these segments do not depend on L shell"? Does it mean that the size of the segments changes a little bit with L, but not enough to be clearly seen when plotted with a logarithmic scale?

Section 3: this section is quite long, would it be possible to add subsection titles to help the reader? You may have a subsection on the protons in the (E,L) space that would start after line 170, one on the helium ions in (E,L) space that would start after line 259, one on the CNO ions in (E,L) space starting after line 287, and finally one on the
protons and helium ions in the (L,B/B0) space starting after line 319.

---

## Author Comment (AC3) · 12 Jul 2019

General comment RC2: The review written by the author and published last year in the journal Space Science Reviews (SSR) concluded that there is a need for the development of radiation belt heavy ion empirical models. The submitted paper gathers available heavy ion measurements obtained all over the space age and tries to pave the way for the development of such models. To do so, invariant parameters (that are constant over a given range of L) are detailed and available measurements are shown on Figures and discussed in detail. The solar cycle variation of heavy ion fluxes is for the first time explored in the submitted article. Finally, the presented measurement

database is important to explore the physical mechanisms that govern the heavy ion radiation belts, what is done by the author in sections 3 and 4. Comparative lessons with what is known for the protons are drawn. There is no doubt that the work presented here is important and may contribute to advances in our understanding and prediction of the heavy ion radiation belts. The submitted article may therefore, according to the reviewer, be ultimately published after several clarifications. The article needs to first be edited for English, in order to make it easily understandable so that it would have an impact on the work of others. There viewer is then wondering: what is new in this article, compared in particular with there view published in SSR last year (see specific comments)? I recommend the article to be revised and reviewed again to see if, after English proof and clarifications, the article would be suitable for publication in Annales Geophysicae.

AC: I am very grateful to the reviewer for very helpful comments on the manuscript. I will try to eliminate my errors in the English language. In this manuscript is a new presentation of experimental data, as well as a comparison of this presentation with the invariants of the spatial-energetic structure of the ERB ion fluxes, and the physical conclusions that follow from these presentations. The comparison and analysis of data on ions with $Z > 1$ obtained in years near the minimum and maximum of solar activity made for the first time. The methods proposed here that allows progress in the problem of constructing empirical models of heavy ions (data for which is clearly not enough) is also a new.

Specific comments

RC2: Section 2: would it be possible to clarify if the invariant parameter values given here come from previous publications or are the outputs of the new study? The values reported here are very important. If the values have now been recomputed or updated, would it be possible to have them highlighted in a table, for instance? A possible use from other researchers would be to compare them with what has been observed by now two orbiters around Jupiter, as these orbiters performed numerous observations

of trapped heavy ions (helium, oxygen, sulfur). Would it be possible to remind, with maybe one sentence, the criterion used to select quiet periods over which the measurements are averaged? The measurements are averaged near solar cycle minimum and near solar cycle maximum. Are the measurements very dispersed around this average in each case or are the standard deviations small compared to the shown averages? A comment may be added in the main text about this. The dataset of heavy ion measurements is limited, but is it large enough to conclude if there is any observable Magnetic Local Time asymmetry in the heavy ion radiation belts, in particular at the lowest considered kinetic energies? Section 4: What are the new conclusions on the physics of the heavy ion radiation belts? If you confirm what has already been reported in previous publications, would it be possible to add a sentence to state it? Otherwise, new findings may be more highlighted in this section and in the conclusion. Lines 453-454: "Here, the experimental database is significantly expanded, many modern measurements of the ion fluxes of the ERB have been added", what are the modern heavy ion measurements added since the article published by the author in 2001? For the protons, one can see the GEO-3 and Van Allen Probe observations, however there does not seem to be any "modern" measurement of heavier ions.

AC: Section 2. The invariants of the ERB structure were obtained in the works of the author 1984-2000 (see references in the manuscript). Of course, I compared the values of these invariants with the results of experiments that were published after 2000, but this had no effect on the average values of the invariants and their variance. When I finished this cycle of works, I looked to the distributions of ions in the belts of other planets (Jupiter, Saturn) and made sure that these belts have such invariants also and their values correspond to the mechanisms discussed in section 4 (for the magnetic fields of these planets). I did not develop this subject, because I had many other interesting and immediate problems. The absence of storms and substorms and Kp < 2-3 were chosen as the criterion for the quiet magnetosphere. The values of scatter of the structure invariants connected mainly with instrumental errors and with the inevitable methodical errors of my analysis (see lines 135-139). For many experiments, the scatter of these

values is much smaller, but by averaging the results, we obtain the ranges of values that are given in lines 122-125 and 128-131. For heavy ions, this scatter is much larger than for protons: there was less data and the instruments had less resolution for them. These invariants are observed in all satellite data in the near-equatorial ERB regions. In my analysis 1984-2001 the solar-cyclic variations were considered in detail only for the last parameter (similarity of the spectra of the different ionic components), which varied greatly during the solar cycle. The invariants connected with the maximum of the spectra and with the intermediate exponential part of the spectra changed significantly less, and the invariants connected with the power-law tail of the spectra were almost independent of solar activity (see lines 140-143). These conclusions are confirmed by Figs. 1-6 and correspond to the mechanisms of formation of these invariants, discussed in Section 4. The results given in section 2 were published only in Russian journals (in the review in SSR and in my articles in Annales Geophysicae they were only mentioned). These results most fully prove the fact of radial diffusion of ions with conservation their first adiabatic invariant in a wide ERB region. Of course, even with a limited set of experimental data, we can conclude that at L > 5-6 even in quiet periods there is a dependency of ion fluxes on MLT. This applies not only to ions of very low energies (ring current) and to the ERB ions also. But for quiet periods on the data of the geosynchronous satellite Gorizont and using averaged empirical model of the magnetic field at the GSO it was established that the dependence of structural invariants on MLT is not here. In connection with these comments, I will revision the text in the Section 2 of the manuscript and try to clarify and expand it. Section 4. Most of the conclusions of Section 4 were published in Russian journals (Kovtyukh, 1999b, 2001). But they are losting in the literature. In addition, these conclusions supplemented. For the first time, the role of fluctuations of the thickness of the plasma sheet of the magnetospheric tail in the formation of the power-law tail of the ion spectra of the ERB is highlighted. These findings will be highlighted in this section and in the Conclusions. Lines 453-454. In my publications of 1984-2001 considered only the ERB region at L > 3. Here I considered a wider range of L (from 1.2 to 8) and E (up to 200 MeV), i.e. considered also the

inner belt. In the general picture included not only the latest data, but also other data obtained at L < 3. Figures 1 and 2 includes data for protons at L < 3 from the satellites Relay-1, Azur, CRRES, Molniya-1, Explorer-45, ISEE-1, OHZORA, ETS-VI, Akebono, GEO-3 and Van Allen Probes. Due to this, we succeeded in tracing the invariants corresponding to the power-law tail of the proton spectra up to L = 1.8. Figures 3 and 4 includes data for helium ions at L < 3 from the satellites OV1-19, Explorer-45, Molniya-2, Prognoz-5, ISEE-1 and Akebono. Figures 5 and 6 includes data for ions of the CNO group at L <3 from the satellites Explorer 45 and ISEE-1. Figures 8 and 9 present data of the satellite Polar for protons and helium ions, which I had not previously considered because of the significant deviation of this satellite's orbit from the equatorial plane. Most of these data refer to areas with B/B0 » 1, where the ERB structure have not invariants. In addition, a main results of this satellite were published after 1999. After the results of these satellites, for the ERB heavy ions with energies above hundreds of keV, unfortunately, nothing new appeared (although many very interesting results were obtained for the ion composition and dynamics of ion fluxes of lower energies, i.e. for the ring current). These explanations will be added to Section 3.

Technical corrections

RC2: The article needs to be edited for English. It would help the reading to explain in the figure captions what the colored lines refer to, even if it is explained in the main text. In the main text, would it be possible to clarify what the maximum deviations shown by the colored vertical segments are: are they based on energy spectra measured by all the satellites, or only a subset? Would it also be possible to clarify the meaning of the following statement "on a logarithmic energy scale, the magnitudes of these segments do not depend on L shell"? Does it mean that the size of the segments changes a little bit with L, but not enough to be clearly seen when plotted with a logarithmic scale? Section 3: this section is quite long, would it be possible to add subsection titles to help the reader? You may have a subsection on the protons in the (E, L) space that would start after line 170, one on the helium ions in (E, L) space that would start after line

259, one on the CNO ions in (E, L) space starting after line 287, and finally one on the protons and helium ions in the (L, B/B0) space starting after line 319.

AC: I will edit and correct the text of the manuscript according to English. Additional explanations in the captions for figures will be added. The maximum deviations of the colored lines correspond to the dispersions of the parameters given in Section 2. For many experiments, especially with heavy ions, the values of these invariants are determined much more accurately not by the spectra, but by the radial profiles of the ion fluxes for different pairs of energy channels. For example, the range L, in which these profiles are parallel to each other, corresponds to the power-law tail of the spectra. On smaller L these profiles begin to converge and intersect with each other; this region corresponds to the exponential part and to the maximum in the spectra. I will clarify this point in the text. All figures presented on a logarithmic scale. For particles moving in the equatorial plane, as in Fig. 1-6, the first adiabatic invariant is E/B(L). The ratios of the upper and lower values for each invariant do not depend on L. Consequently, the difference of the logarithms of these values is also independent on L. Therefore, the vertical segments on the colored lines can be shifted along the corresponding lines without changing their sizes on the energy scale. In other words, "on a logarithmic energy scale the magnitudes of these segments do not depend on L shell". This is also true for magnetic traps of non-dipole type, i.e., in our case, for large L. Invariants corresponding to index of the power-law tail and similarity of the distributions of various ionic components cannot be plotted in Fig. 1-6 by color lines: they do not depend on E and L. First from these invariants manifests itself in a parallel course of isolines on sufficiently large L and is calculated over the intervals between these isolines. Second from these invariants is manifested only if overlapped Figs. 1-3-5 and also Figs. 2-4-6 (in this case, it is necessary leave only the isolines and give them a different color or thickness). With such an overlay, one can see that on L > 3.5, the structure of isolines in even figures (minimum solar activity) is closer to each other than on odd figures (maximum solar activity). Since on the figures energy is presented in MeV/n, this means that at the minimum of solar activity the similarity parameter of these

distributions is closer to Mi (as in the heliosphere), and at the maximum of solar activity this is closer to Qi (as in the ring current). From the experimental spectra and the radial profiles of the fluxes ratios for different ion components this parameter calculated more precisely. I will supplement these remarks in Section 3 of the manuscript. I agree to break up Section 3 into subsections for different ionic components.

———————————————————

---

## Author Comment (AC4) · 12 Jul 2019

General comments

This paper presents a parametric model for the density of various ion species in the Earth radiation belts. This model describes the global structure of the radiation belts for protons, helium, and for ions of the CNO group. Based on extensive satellite data, the parameters of the models have been fitted, independently for the proton populations and for the other ions. The validity of the model is discussed, species by species, by comparison with in-flight data. The solar cycle dependency is presented. Finally, a physical interpretation of the model is detailed.

Overall, the paper is difficult to understand. Firstly, the English is very poor (see technical corrections hereafter), and difficult to grasp. Secondly, the model is not appropriately explained, as the author refers to previously published literature, which could not be accessed by this reviewer. Thirdly, as described in the specific comments, the figures are not very clear and supportive of the arguments developed in this paper.

I am very grateful to the Referee for these comments: they help clarify the essence of my work and I, of course, will make all the necessary changes in the revised manuscript.

In many places of this manuscript, starting with the title, the word "model" is found and in almost all cases this word has the most general meaning here: any observation or measurement is a physical model (modeled by instruments or a brain). But in fact, no empirical or mathematical models are used in this manuscript, and what is presented in Fig. 1-9, these are collections of experimental data.

Associations with models (in the narrow sense) also invoke the word "parameters".

What presented here, in section 2, and called the invariant parameters of ERB are not the parameters of mathematical or empirical models of ERB. These parameters obtained directly from the results of various experiments, from the figures of the corresponding articles. They were found for each energy spectrum and for each dependence from $L$ of the ions fluxes, and only then these parameters were averaged (separately for each ion component).

These invariants exist only in the region where the transport (radial diffusion) of ions dominates their losses. With increasing $B/B_0$, the rate of radial diffusion of ions decreases, and the rate of their loss increases rapidly. Therefore, on small $L$ and for large $B/B_0$, these invariants are not applicable to ERB and in Figs. 7-9 they are not presented (these figures are given only for completeness).

Author's papers in journals Cosmic Res. and Geomagn. Aeron. (in English) can be found in any major university library. But in order to make sure that there are such invariants and in the correctness of the values given here, it is not necessary to read these articles. It is enough to open several articles where there are experimental spectra or radial profiles of ion fluxes with $E > 0.1$ MeV for $L > 3$, obtained in quiet periods near the equatorial plane.

I will take into account all these remarks and the words "model" and "parameter" will be saved only where it says about specific ERB models.

I checked the descriptions of the figures and added them with the necessary explanations.
I agree that I don't speak English well enough and gratefully accept any corrections to the text of the manuscript.

Specific comments

Section 2 of this paper presents the model parameters and their measured values, but not the model itself, which is only suggested by the description of the parameters. A detailed and self-sufficient description of the model should be given in this section.

On section 3, numerous similar figures are presented. It is not clear how these figures were obtained from the data. In particular, these figures present iso-lines on power of tens, with most satellite data points placed on the iso-lines, which suggest some interpolation was done on the satellite data. Section 3 should detail how this figures were made.

The conclusions of the comparison of figures 1 to 4 proposed at line 269 are not clearly apparent in the figures. Similarly, the low-altitude effects described at line 352 cannot be clearly seen on the figures, because the transformation B/Beq to altitude is not straightforward (for instance, the 1000 km altitude line could be drawn on figure 7-9 to support the arguments of this paragraph).

A reference should be provided at line 347 for the dependency of the radial diffusion rate on B/Beq.

A figure supporting the information at line 362-364 about the CNO group data could be provided.

The invariants presented in Section 2 are not tied to any particular model, but they can be used in works on the creation of both empirical and mathematical models of spatial-energy distributions of ERB ions. In particular, they were tested in many my works on modeling the pitch-angle distributions of protons, the empirical model of ion fluxes for region of the geosynchronous orbit (GSO), the daily course of ion fluxes on the GSO and in other problems.

Points on fig. 1-9 were obtained from experimental radial profiles of differential ion fluxes. It was taken into account that, for different authors, these fluxes have different dimensions. For example, for ions with $Z > 1$, these fluxes are given in $(\text{cm}^2 \text{ s ster MeV/n})^{-1}$ or in $(\text{cm}^2 \text{ s ster MeV})^{-1}$; in the latter case the same ion flux will have a value of $Z$ times less.

Iso-lines in Fig. 1-9 are envelopes of experimental points (they are constructed by the method of the $\chi^2$).

It was important to choose a form of representation (space of variables) in which the results of different experiments (with different sets of the energy chennels) one could accommodate organically and without resorting to interpolations and extrapolations of the data. As such representation was chosen the space $\{E. L\}$. All other representations, such as radial profiles of ion fluxes for different energies, suggest interpolation and extrapolation of an experimental data; this is greatly complicates the solution of our problem and introduces large systematic errors and arbitrariness in the choice of curves approximation.

To present the data obtained outside the equatorial plane, it was natural to use the space $\{B/B_0, L\}$ for ions of different energies. Here I had to resort to interpolation of data and Figs. 7–9 are less complete and accurate compared to Figs. 1–6.

For comparison Figs. 1-2 with Figs. 3-4, specific values of proton and helium ion fluxes are given (lines 269-271).

For Figs. 7-9 interpretation can be simplified, and then it is not necessary to put on these figures the dependences $B/B_0(L)$ for fixed heights.

For the dependency of the radial diffusion rate on $B/B_0$ a reference are added (at line 347).

There are not enough data for CNO group ions and they are very fragmentary; build or them figures like to Fig. 7-9 while is impossible. One can only make assumptions on the basis of available data and general considerations. I changed the sentence at lines 362-364, made it more careful.

Technical corrections

Numerous English errors have been found, for instance on lines 21, 24, 34, 48, 63, 67, 69, 79, 118, 132, 134, 137, 161, 163, 166, 173, 185, 193, 201, 215, 220, 226, 239,271, 290, 314, 338, 379, 387, 389, 398, 403, 469.

In the figure legends, the MeV unit is displayed in Cyrillic. Moreover, the model lines on the figures are not described in the legends.

I would correct the errors in the revised version of the manuscript.

All dimensions of the MeV unit displayed in Cyrillic replaced by on Latino.

In the caption to Figs. 1-6 added explanations to color lines. In Figs. 1–6, the color lines are show dependencies on $L$ of the ion energies corresponding to the structure invariants ($E \propto \mu L^{-3}$) and also the maximum energy of ions, which can be trapped in the ERB($E \propto L^{-4}$).

---

## Author Comment (AC6) · 17 Aug 2019

**Final author comments on the manuscript "Invariants of the Spatial-Energy Structure and Modeling of the Earth's Ion Radiation Belts" by Alexander S. Kovtyukh for Anonymous Referee #1**

I am very grateful to Referee #1 for very helpful comments: they help clarify the essence of my work. All of them are taken into account in the revised manuscript.

General comments

(1) This paper presents a parametric model for the density of various ion species in the Earth radiation belts. This model describes the global structure of the radiation belts for protons, helium, and for ions of the CNO group. Based on extensive satellite data, the parameters of the models have been fitted, independently for the proton populations and for the other ions. The validity of the model is discussed, species by species, by comparison with in-flight data. The solar cycle dependency is presented. Finally, a physical interpretation of the model is detailed.

(2) In many places of this manuscript, starting with the title, the word "model" is found and in almost all cases this word has the most general meaning; no empirical or mathematical models are used in this manuscript and in Fig. 1-9 are presented experimental data. Associations with models also invoke the word "parameters". But the invariant parameters of ERB (presented in Section 2 and in Figs. 1-6) are not the parameters of mathematical or empirical models. These parameters obtained directly from the results of various experiments. Model (theoretical) calculations were made only to the maximum energy of ions, which can be trapped in the ERB (green line in Figs. 1-6).

(3) More detailed definitions of all quantities considered here are given (lines 97-102, 138-147, 159-160, 201-218, 363-367, 471-477). The words "model" and "parameter" conserved only where it comes to specific ERB models.

(1) Overall, the paper is difficult to understand. Firstly, the English is very poor (see technical corrections hereafter), and difficult to grasp.

(2) I agree that I don't speak English well enough and I corrected the text of the manuscript.

(3) I edited the manuscript. Many words and sentences are clarified. New paragraphs added and some paragraphs moved.

(1) Secondly, the model is not appropriately explained, as the author refers to previously published literature, which could not be accessed by this reviewer.

(2) Empirical or mathematical models of the ERB are not used in this manuscript. There made only a systematization of the experimental data and is considered how to use the results obtained in this paper for modeling of ion fluxes in the ERB.

(3) All necessary explanations are given in the manuscript (lines 97-102, 138-147, 159-160, 201-218). The words "model" and "parameter" conserved only where are mentioned a specific ERB models.

(1) Thirdly, as described in the specific comments, the figures are not very clear and

supportive of the arguments developed in this paper.

> (2) The descriptions of the figures are supplemented with the necessary explanations (in the text and in the captions of the figures).
> (3) I added the necessary explanations to the figures (lines 191-192, 199-202, 205-214).

Specific comments

(1) Section 2 of this paper presents the model parameters and their measured values, but not the model itself, which is only suggested by the description of the parameters. A detailed and self-sufficient description of the model should be given in this section.

> (2) The invariants presented in Section 2 are not connected to any particular model, but they can be used in works on the creation of both empirical and mathematical models of spatial-energy distributions of the ERB ions.
> (3) Detailed definitions of all quantities considered here are given (lines 97-102, 138-147, 159-160, 201-218). The words "model" and "parameter" conserved only where it comes to specific ERB models.

(1) On section 3, numerous similar figures are presented. It is not clear how these figures were obtained from the data. In particular, these figures present iso-lines on power of tens, with most satellite data points placed on the iso-lines, which suggest some interpolation was done on the satellite data. Section 3 should detail how this figures were made.

> (2) Points on Figs. 1-9 were obtained from the experimental radial profiles of differential ion fluxes. Iso-lines in Figs. 1-9 are envelopes of the experimental points. Figs. 1-6 were constructed without using the interpolations and extrapolations of the experimental data. Figs. 7-9 were constructed with using the interpolation and extrapolation of the data.
> (3) The choice of representations of the experimental data and methods of constructing of Figs. 1-9 are explained in details (lines 172-181, 191-192, 205-218, 363-367).

(1) The conclusions of the comparison of figures 1 to 4 proposed at line 269 are not clearly apparent in the figures.

> (2) For comparison Figs. 1-2 with Figs. 3-4 and with Figs. 5-6 are given concrete values of the fluxes of protons and helium ions (lines 306-309) and fluxes of protons and CNO group ions (lines 347-352).
> (3) A more accurate formulation of the conclusions about stronger solar-cyclic variations of heavy ion fluxes compared to proton fluxes is given (lines 306-309, 353-354, 499-500).

(1) Similarly, the low-altitude effects described at line 352 cannot be clearly seen on the figures, because the transformation B/Beq to altitude is not straightforward (for instance, the 1000 km altitude line could be drawn on figure 7-9 to support the arguments of this paragraph).

(2) For Figs. 7-9 my interpretation were simplified, and after that it is not necessary to put on these figures the dependences $B/B_0(L)$ for fixed heights.
(3) A simpler interpretation of Figs. 7-9 is given (lines 396-398).

A reference should be provided at line 347 for the dependency of the radial diffusion rate on B/Beq.
(2) For the dependency of the radial diffusion rate on $B/B_0$ a reference are added (at line 395).
(3) I made it (line 395).

A figure supporting the information at line 362-364 about the CNO group data could be provided.
(2) There are not enough data for CNO group ions and they are very fragmentary. Constructing for them figures like to Fig. 7-9 while is impossible. But one can made assumptions on the basis of available data and general considerations.
(3) I clarified the relevant conclusion (lines 400-407).

Technical corrections

(1) Numerous English errors have been found, for instance on lines 21, 24, 34, 48, 63, 67, 69, 79, 118, 132, 134, 137, 161, 163, 166, 173, 185, 193, 201, 215, 220, 226, 239,271, 290, 314, 338, 379, 387, 389, 398, 403, 469.
(2) I gratefully accept these corrections to the text of the manuscript.
(3) I corrected all these errors.

(1) In the figure legends, the MeV unit is displayed in Cyrillic.
(2-3) I corrected all these errors.

(1) Moreover, the model lines on the figures are not described in the legends.
(2) I describe all lines which are shown in Fig. 1-6.
(3) I made these additions (lines 719-721, 725-727, 731-733, 737-739, 743-745, 749-751).

I am very grateful to Referee #1 for very helpful comments.

With grand regard,
Alexander S. Kovtyukh

---

## Author Comment (AC7) · 18 Aug 2019

**Final author comments on the manuscript "Invariants of the Spatial-Energy Structure and Modeling of the Earth's Ion Radiation Belts" by Alexander S. Kovtyukh for Anonymous Referee #2**

I am very grateful to Referee #2 for very helpful comments on the manuscript. All these comments are taken into account in the revised manuscript.

General comments

The review written by the author and published last year in the journal Space Science Reviews (SSR) concluded that there is a need for the development of radiation belt heavy ion empirical models. The submitted paper gathers available heavy ion measurements obtained all over the space age and tries to pave the way for the development of such models. To do so, invariant parameters (that are constant over a given range of $L$) are detailed and available measurements are shown on Figures and discussed in detail. The solar cycle variation of heavy ion fluxes is for the first time explored in the submitted article. Finally, the presented measurement database is important to explore the physical mechanisms that govern the heavy ion radiation belts, what is done by the author in sections 3 and 4. Comparative lessons with what is known for the protons are drawn.

There is no doubt that the work presented here is important and may contribute to advances in our understanding and prediction of the heavy ion radiation belts. The submitted article may therefore, according to the reviewer, be ultimately published after several clarifications.

(1) The article needs to first be edited for English, in order to make it easily understandable so that it would have an impact on the work of others.

(2) I corrected the text of the manuscript.
(3) I edited the manuscript. Many words and sentences are clarified. New paragraphs added and some paragraphs moved.

(1) There viewer is then wondering: what is new in this article, compared in particular with there view published in SSR last year (see specific comments)?

(2) A new view of known experimental data. There are considered a wider ranges of $L$ and $E$ and not only the outer, but also the inner belt. The comparison and analysis of data on ions with $Z > 1$ obtained in years near the minima and maxima of solar activity made for the first time. New methods proposed here allows progress in the problem of constructing empirical models of heavy ions (data for which is clearly not enough).
(3) New results obtained in this work are highlighted (lines 14-27, 172-181, 199-200, 306-309, 247-354, 471-477, 479-484, 489-502, and Figs. 1-9).

(1) I recommend the article to be revised and reviewed again to see if, after English proof and clarifications, the article would be suitable for publication in Annales Geophysicae.

(2-3) Manuscript is completely revised. Many details of this work are clarified.

Specific comments

(1) Section 2: would it be possible to clarify if the invariant parameter values given here come from previous publications or are the outputs of the new study? The values reported here are very important. If the values have now been recomputed or updated, would it be possible to have them highlighted in a table, for instance?

(2) The invariants of the ERB structure were obtained for $L > 3$ in the works of the author 1984-1999 (see references in the manuscript). Here are considered more complete data of the satellites and for protons with $E > 10$ MeV invariants $\mu_b$ and $\gamma$ were traced to $L \sim 2$. Of course, I compared the values of these invariants with the results of experiments that were published after 2000, but this had no effect on the average values of the invariants and their variances. From the experimental radial profiles of the fluxes for different ion energy the values of these invariants were obtained most precisely.

(3) I revised Section 2, clarify and expand it. It is explained how the invariants of the structure of the ERB ion fluxes were obtained (lines 97-102, 118-119, 138-147, 165-170, 288-295, 317-318, 337-346).

(1) A possible use from other researchers would be to compare them with what has been observed by now two orbiters around Jupiter, as these orbiters performed numerous observations of trapped heavy ions (helium, oxygen, sulfur).

(2) The distributions of ions in the belts of other planets (Jupiter, Saturn) have such invariants also and their values correspond to the mechanisms discussed in Section 5 (for the magnetic fields of these planets).

(3) Remark about radiation belts of Jupiter and Saturn is given (lines 471-477).

(1) Would it be possible to remind, with maybe one sentence, the criterion used to select quiet periods over which the measurements are averaged?

(2) The absence of storms and substorms and $Kp < 2$-3 were chosen as the criterion for the quiet magnetosphere.

(3) The criterion of the quiet magnetosphere given (lines 138-139, 190-191, 226, 297, 326).

(1) The measurements are averaged near solar cycle minimum and near solar cycle maximum. Are the measurements very dispersed around this average in each case or are the standard deviations small compared to the shown averages? A comment may be added in the main text about this.

(2) The values of scatter of the structure invariants connected mainly with instrumental errors and with the inevitable methodical errors. For many experiments, the scatter of these values is much smaller the average results given in Section 2. In my analysis 1984-2001 the solar-cyclic variations were considered in detail only for the parameter $\xi_i$, which varied greatly during the solar cycle.

(3) Questions related to the scatter of the values of structural invariants are clarified (lines 155-160). Questions related to solar-cyclic variations in the values of structural invariants are discussed (lines 161-170, 239-240).

(1) The dataset of heavy ion measurements is limited, but is it large enough to conclude if there is any observable Magnetic Local Time asymmetry in the heavy ion radiation belts, in particular at the lowest considered kinetic energies?

(2) At $L > 5$-6 even in quiet periods there is a dependency of ion fluxes on MLT. But for quiet periods on the data of the geosynchronous satellite Gorizont (and using averaged empirical model of the magnetic field at the GSO) it was established that in this region no the dependence of structural invariants on MLT.

(3) The question of the dependence of ion fluxes and structural invariants of the ERB on MLT at $L > 5$–6 is discussed (lines 215-218).

(1) Section 4: What are the new conclusions on the physics of the heavy ion radiation belts? If you confirm what has already been reported in previous publications, would it be possible to add a sentence to state it? Otherwise, new findings may be more highlighted in this section and in the conclusion.

(2) For the first time, the role of fluctuations of the thickness of the plasma sheet of the magnetospheric tail in the formation of the power-law tail of the ion spectra of the ERB is highlighted. A hypothesis about analogous structure invariants of the radiation belts of other planets is suggested.

(3) These findings are highlighted in Section 5 and in the Conclusion (lines 448-449, 466, 471-477, 505-509).

(1) Lines 453-454: "Here, the experimental database is significantly expanded, many modern measurements of the ion fluxes of the ERB have been added", what are the modern heavy ion measurements added since the article published by the author in 2001? For the protons, one can see the GEO-3 and Van Allen Probe observations, however there does not seem to be any "modern" measurement of heavier ions.

(2) In my publications of 1984-2001 considered only region of the ERB at $L > 3$ and ions with $E < 10$ MeV. Here I considered a wider range of $L$ (from 1.2 to 8) and $E$ (up to 200 MeV), i.e. is considered also the inner belt. Due to this, we succeeded in tracing the invariants corresponding to the power-law tail of the proton spectra up to $L \sim 2$. Figures 8 and 9 present data of the satellite Polar for protons and helium ions, which I had not previously considered because of the significant deviation of this satellite's orbit from the equatorial plane.

(3) I describing what additional experimental results, compared with my publications of 1984-2001, were used in this work (lines 172-173, 288-295, 363-367).

Technical corrections

(1) The article needs to be edited for English.

(2) I corrected the text of the manuscript.

(3) I edited the manuscript.

(1) It would help the reading to explain in the figure captions what the colored lines refer to, even if it is explained in the main text.

(2) Additional explanations in the captions for Figs. 1-6 are added.

(3) It is made (lines 718-721, 724-727, 731-733, 737-739, 743-745, 749-751).

(1) In the main text, would it be possible to clarify what the maximum deviations shown by the colored vertical segments are: are they based on energy spectra measured by all the satellites, or only a subset?

(2) The maximum deviations of the colored lines correspond to the dispersions of the parameters given in Section 2. For many experiments, especially with heavy ions, the values of these invariants are determined much more accurately not by the spectra, but by the radial profiles of the ion fluxes for different pairs of energy channels.

(3) It is made (lines 138-147, 159-160, 191-192, 201-214).

(1) Would it also be possible to clarify the meaning of the following statement "on a logarithmic energy scale, the magnitudes of these segments do not depend on $L$ shell"? Does it mean that the size of the segments changes a little bit with $L$, but not enough to be clearly seen when plotted with a logarithmic scale?

(2) For particles moving in the equatorial plane, as in Fig. 1-6, the first adiabatic invariant is $E/B(L)$. The ratios of the upper and lower values for each invariant do not depend on $L$. Consequently, the difference of the logarithms of these values is also independent on $L$. Therefore, the vertical segments on the colored lines can be shifted along the corresponding lines without changing their sizes on the energy scale. This is also true for magnetic traps of non-dipole type, i.e., in our case, for large $L$.

(3) It is explained (lines 205-209).

(1) Section 3: this section is quite long, would it be possible to add subsection titles to help the reader? You may have a subsection on the protons in the (E, L) space that would start after line 170, one on the helium ions in (E, L) space that would start after line 259, one on the CNO ions in (E, L) space starting after line 287, and finally one on the protons and helium ions in the (L, B/B0) space starting after line 319.

(2) Section 3 is divided into subsections (for different ion components). Figures 7-9 and the associated text is detached in Section 4.

(3) As a result, the number of sections increased from 5 to 6 (lines 171, 225, 296, 325, 362).

I am very grateful to Referee #2 for very helpful comments.

With grand regard,
Alexander S. Kovtyukh